# Modeling flexible behavior in childhood to adulthood shows age-dependent learning mechanisms and less optimal learning in autism in each age group

Daisy Crawley[1‡]*, Lei Zhang[2,3,4‡], Emily J. H. Jones[5], Jumana Ahmad[1,6], Bethany Oakley[1], Antonia San José Cáceres[1,7], Tony Charman[8,9], Jan K. Buitelaar[10,11,12], Declan G. M. Murphy[1,9,13], Christopher Chatham[4], Hanneke den Ouden[10‡], Eva Loth[1,13‡], the EU-AIMS LEAP group[¶]

1 Department of Forensic and Neurodevelopmental Sciences, Institute of Psychiatry, Psychology & Neuroscience, King's College London, London, United Kingdom, 2 Institute of Systems Neuroscience, University Medical Center Hamburg-Eppendorf, Hamburg, Germany, 3 Neuropsychopharmacology and Biopsychology Unit, Department of Cognition, Emotion, and Methods in Psychology, Faculty of Psychology, University of Vienna, Vienna, Austria, 4 F. Hoffmann La Roche, Innovation Center Basel, Basel, Switzerland, 5 Centre for Brain and Cognitive Development, Birkbeck, University of London, London, United Kingdom, 6 Department of Psychology, Social Work and Counselling, University of Greenwich, London, United Kingdom, 7 Instituto de Investigación Sanitaria Gregorio Marañón, Departamento de Psiquiatría del Niño y del Adolescente, Hospital General Universitario Gregorio Marañón, Madrid, Spain, 8 Department of Psychology, Institute of Psychiatry, Psychology & Neuroscience, King's College London, London, United Kingdom, 9 South London and Maudsley NHS Foundation Trust (SLaM), London, United Kingdom, 10 Donders Institute for Brain, Cognition and Behaviour, Centre for Cognitive Neuroimaging, Radboud University, Nijmegen, the Netherlands, 11 Department of Cognitive Neuroscience, Radboud University Nijmegen Medical Center, Nijmegen, the Netherlands, 12 Karakter Child and Adolescent Psychiatry University Centre, Nijmegen, the Netherlands, 13 Sackler Institute for Translational Neurodevelopment, Institute of Psychiatry, Psychology & Neuroscience, King's College London, London, United Kingdom

‡ DC and LZ are joint first authors on this work. HdO and EL are joint senior authors on this work.
¶ Membership of the EU-AIMS LEAP group is provided in the Acknowledgments.
* daisy.crawley@kcl.ac.uk

**Data Availability Statement:** The underlying numerical data for each figure within this paper can be found in the Supporting Information files. The

## Abstract

Flexible behavior is critical for everyday decision-making and has been implicated in restricted, repetitive behaviors (RRB) in autism spectrum disorder (ASD). However, how flexible behavior changes developmentally in ASD remains largely unknown. Here, we used a developmental approach and examined flexible behavior on a probabilistic reversal learning task in 572 children, adolescents, and adults (ASD $N$ = 321; typical development [TD] $N$ = 251). Using computational modeling, we quantified latent variables that index mechanisms underlying perseveration and feedback sensitivity. We then assessed these variables in relation to diagnosis, developmental stage, core autism symptomatology, and associated psychiatric symptoms. Autistic individuals showed on average more perseveration and less feedback sensitivity than TD individuals, and, across cases and controls, older age groups showed more feedback sensitivity than younger age groups. Computational modeling revealed that dominant learning mechanisms underpinning flexible behavior differed across developmental stages and reduced flexible behavior in ASD was driven by less optimal learning on average within each age group. In autistic children, perseverative errors were

raw data and code are available upon request from the EU-AIMS LEAP group via the corresponding author.

**Funding:** This work was supported by funding from EU-AIMS, AIMS-2 TRIALS, the MRC UK, and the National Institute for Health Research (NIHR) Biomedical Research Centre at South London and Maudsley NHS Foundation Trust and King's College London. EU-AIMS receives support from the Innovative Medicines Initiative (IMI) Joint Undertaking (JU) under grant agreement no. 115300, the resources of which are composed of financial contributions from the European Union's Seventh Framework Programme (grant FP7/2007-2013), from the European Federation of Pharmaceutical Industries and Associations companies' in-kind contributions and from Autism Speaks. AIMS-2 TRIALS received funding from the IMI 2 JU under grant agreement no. 777394, with support from the European Union's Horizon 2020 research and innovation program and EFPIA, Autism Speaks, Autistica, SFARI, and the Simons Foundation. LZ was supported by the Research Promotion Fund (FFM) for young scientists of the University Medical Center Hamburg-Eppendorf and Vienna Science and Technology Fund (WWTF VRG13-007). The funders had no role in study design, data collection and analysis, decision to publish, or preparation of the manuscript.

**Competing interests:** I have read the journal's policy and the authors of this manuscript have the following competing interests: ASJC is a consultant for Servier Laboratories and is involved in clinical trials conducted by Servier. The present work is not related to this relationship. JKB has been a consultant to/member of advisory board and/or speaker for Janssen Cilag BV, Eli Lilly, Lundbeck, Shire, F. Hoffman-La Roche, Novartis, Medice, and Servier. CC is a full-time employee of F. Hoffmann La Roche. TC has received research grant support from the Medical Research Council (UK), the National Institute for Health Research, Horizon 20202 and the Innovative Medicines Initiative (European Commission), MQ, Autistica, FP7 (European Commission), the Charles Hawkins Fund, and the Waterloo Foundation. He has served as a consultant to F. Hoffmann-La Roche. He has received royalties from Sage Publications and Guilford Publications. DGMM sits on the Scientific Advisory Board for F. Hoffmann-La Roche and receives an honorarium. The present work is not related to this relationship. There are no other declarations of interest.

**Abbreviations:** ADHD, attention-deficit hyperactivity disorder; ADI-R, Autism Diagnostic Interview-Revised; ASD, autism spectrum disorder;

positively related to anxiety symptoms, and in autistic adults, perseveration (indexed by both task errors and model parameter estimates) was positively related to RRB. These findings provide novel insights into reduced flexible behavior in relation to clinical symptoms in ASD.

## Introduction

Flexible behavior is a fundamental part of everyday life. It requires learning from feedback to guide decisions and adapting responses when feedback changes. These cognitive processes are implicated in a range of neurodevelopmental and neuropsychiatric conditions, including autism spectrum disorder (ASD; [1]), as well as attention-deficit hyperactivity disorder (ADHD) and anxiety, both of which frequently co-occur in ASD [2–5]. In particular, reduced flexible behavior is suggested to underpin core features of restricted, repetitive behaviors (RRB) in ASD, such as insistence on sameness. However, current evidence is inconclusive, and the mechanisms by which these impairments arise remain unclear [6, 7]. Studies of neurotypical individuals show that the cognitive processes underlying flexible behavior and reinforcement learning change through childhood and adolescence into adulthood [8, 9]. Therefore, a developmental approach within ASD that characterizes component learning processes is likely to bring us closer to understanding mechanisms of (in)flexible behavior and identifying therapeutic targets.

Probabilistic reversal learning (PRL) paradigms require individuals to find a balance between learning structure in an uncertain environment while remaining flexible to change [10]. Typically, participants must learn using feedback which of a set of stimuli is most rewarded and adapt their responses when the rule changes, in order to maximize favorable outcomes. PRL paradigms therefore provide a direct assessment of flexible choice behavior (in addition to tapping reinforcement learning), as they require information to be integrated over a number of trials in order to detect true changes, and—much like interacting with our environment—this trial-and-error learning is continually updated throughout the task. Furthermore, PRL paradigms do not require tracking of extradimensional shifts, thereby constraining the recruitment of additional cognitive domains [11, 12].

Previous literature has reported reduced reversal learning in ASD relative to controls and a positive relationship between reversal errors and RRB [1, 13]. In contrast, others have reported poorer overall task performance but unspecific to reversal adaptation [14, 15], or no differences in reversal learning nor any associations with ASD symptomatology [16, 17]. It is worth noting that these inconsistencies in ASD-related changes in cognitive flexibility are also reflected in the broader literature using alternative paradigms (see [7, 18] for reviews).

With respect to reinforcement learning, studies of reward processing suggest atypical or diminished neural responses to rewards in ASD [19–22], though results from adolescent studies are less consistent [23–25]. If reinforcement is differentially experienced in ASD, it is likely to impact on decision-making processes and behavior. In addition to establishing differences, associations between learning and phenotypic correlates warrant further study in order to elucidate whether such differences necessarily manifest in impairments related to symptom severity.

Several factors may have contributed to inconsistencies in the literature. First, previous studies have often studied single age groups or a broad age range within a small sample size. Evidence from both cognitive and neuroimaging studies attests to important developmental differences in reinforcement learning and flexible behavior in neurotypical individuals [26–28]. Young children often perseverate, taking longer than older children to learn new rules

BAI, Beck Anxiety Inventory; BYI-II, Beck Youth Inventories–Second Edition; CI, confidence interval; CU, counterfactual update; EWA-DL, experience-weighted attraction–dynamic learning rate model; IU, intolerance of uncertainty; PRL, probabilistic reversal learning; RBS-R, Repetitive Behavior Scale-Revised; RDoC, research domain criteria framework; RL, reinforcement learning; R-P, reward-punishment; RRB, restricted, repetitive behaviors; RW, Rescorla-Wagner; SRS-2 SCI, Social Responsiveness Scale—2nd Edition Social Communication Index; TD, typical development.

and switch their responses [8]. During adolescence, notable changes in goal-directed decision-making occur, often manifesting in risky decisions thought to be attributable to hypersensitivity to rewards [29–31]. In adulthood, there is evidence for the use of more sophisticated, "controlled" cognitive strategies [32, 33]. Hence, a developmental approach in ASD is needed to ascertain whether potential impairments reflect delayed development or atypical cognitive processes.

Second, previous studies have also tended to use task performance measures that often aggregate error scores and do not directly characterize learning processes governing behavior. Computational models capture the dynamics of learning over time—emulating a participant's experience—and delineate component processes underlying PRL by approximating mechanisms that may have led to task behavior. Estimating and comparing different reinforcement learning models allows for the evaluation of competing mechanisms by quantifying how likely each model is to have generated the observed behavior. Moreover, by approximating putative mechanisms, computational models enable better mapping between behavior and neurobiology, particularly important for understanding neurodevelopmental disorders [34].

Studies of ASD using modeling have shown evidence of slower, faster, and equal rates of learning compared to neurotypical individuals. Optimal learning rates depend on the stability of the task environment. A changeable environment requires fast learning guided by recent feedback, whereas a stable environment requires slower learning over time (e.g., [35, 36]). Crucially, probabilistic feedback also requires learning to ignore "misleading" punishment. Previously, autistic adults were shown to have a slower learning rate than neurotypical adults when using higher-probability reward contingencies, but they performed comparably or outperformed neurotypical adults when the contingency was near chance [21, 22]. Perhaps, then, a key difficulty lies in learning regularities and ignoring irregularities, in addition to learning change per se [37]. This is consistent with previous findings of a tendency to "overlearn" volatility in ASD adults, resulting in reduced learning of probabilistic errors [38]. Whether these findings extend to children and adolescents (see [39] for differing findings) and which underlying processes are different in ASD remain to be seen.

Here, we examined learning processes underlying flexible behavior in ASD and typical development (TD) across developmental stages using a PRL paradigm. Our secondary aim was to investigate possible relationships with symptomatology in ASD. To achieve this, we (1) tested a large sample of individuals with a wide age range that was sufficiently powered to compare children, adolescents, and adults and (2) used reinforcement learning models to compare quantitative mechanistic explanations of flexible behavior and identify the latent processes on which individuals may differ. We included measures of RRB subtypes as our focus, social-communication difficulties for comparison, and associated symptoms of ADHD and anxiety as frequently co-occurring features that may also relate to atypical learning and flexible behavior. Based on previous literature, we hypothesized that younger age groups would perform less well on the task than older age groups and that autistic individuals would perform less well than neurotypical individuals. Additionally, we hypothesized differences in dominant underlying cognitive processes across development. Finally, we predicted that reduced flexible behavior would be related to higher RRB symptom severity, in particular behavioral rigidity/insistence on sameness.

## Methods

### Ethics statement

The study was approved by the independent local ethics committees of the participating centers (London Queen Square Health Research, Authority Research Ethics Committee: 13/LO/

56; Radboud University Medical Centre Institute Ensuring Quality and Safety Committee on Research Involving Human Subjects Arnhem-Nijmegen: 2013/455; UMM University Medical Mannheim, Medical Ethics Commission II: 2014–540 N-MA; University Campus Bio-Medical Ethics Committee of Rome: 18/14 PAR ComET CBM) and conducted according to the principles expressed in the Declaration of Helsinki. Written informed consent was obtained from all participants and/or their parent/guardian (when appropriate) prior to the study.

## Participants

This study was part of the EU-AIMS Longitudinal European Autism Project (LEAP; [40, 41])—a multidisciplinary, multicenter study of children (6–11 years), adolescents (12–17 years), and adults (18–30 years) with and without ASD from six European sites. The current study included data from 321 individuals with an existing clinical diagnosis of ASD and 251 typically developing (TD) individuals, with full-scale IQ scores ranging from 74 to 148. Descriptive statistics for the sample are listed in Table 1. Full-scale IQ was measured using the Wechsler scales (see [41]). Although ASD individuals were additionally assessed using the Autism Diagnostic Observation Schedule [42, 43] and Autism Diagnostic Interview-Revised (ADI-R, [44]), reaching instrument cutoffs were not inclusion criteria, as clinical judgment has been found to consistently improve diagnostic stability [45]. However, task behavioral analyses were repeated in a subset of individuals who meet ADI-R criteria as specified by [46] (S1 Table). Although the full EU-AIMS LEAP sample includes individuals with mild intellectual disabilities ($N = 83$), initial analyses showed evidence of poor task learning in this group, and thus they were omitted from further analyses. Those with only partial data ($N = 3$) or who chose the same stimulus throughout the task ($N = 1$) were excluded from analysis (see S1 Text for further sample information).

## Experimental paradigm

Participants completed a computerized PRL task whereby they were instructed to choose one of two colored shapes (vertical yellow bars or horizontal blue bars) presented in two of four possible locations with an 80:20 reward/punishment contingency (Fig 1A). Positive feedback consisted of green, smiling emoticons and negative feedback of red, frowning emoticons (i.e., reward/punishment) and accompanying sounds (bell chime/buzzer, respectively). The task employed a pseudorandom fixed sequence comprising 80 trials with a reversal midway. Participants' first stimulus choice was considered correct in the acquisition phase; after the reversal, the initially incorrect stimulus became the usually rewarded stimulus and vice versa (Fig 1B and 1C). To reduce task demand and avoid potential floor effects in the younger age groups or clinical sample, the contingency ratio was higher than some previous studies (70:30; [10, 47]). Participants used arrow keys to respond and had unlimited response time per trial (see S1 Text for task instructions). This paradigm has previously been used in neurotypical individuals and other clinical groups [47, 48] and was specified by the European Medicines Agency in their letter of support for EU-AIMS LEAP [49].

## Analysis of task behavior

Behavioral performance on the task was assessed using accuracy during acquisition and reversal phases, perseverative errors, and win/lose feedback sensitivity. Accuracy was quantified as the proportion of correct responses. Perseverative errors were defined as two or more consecutive errors during the reversal phase—i.e., trials in which the participant chose the previously rewarded stimulus, despite negative feedback—and are reported as a proportion of reversal phase trials. Win-stay and lose-shift behaviors index the effect of an outcome on the subsequent choice. They are defined, respectively, as repeating the previous choice following reward

**Table 1. Participant characteristics (total *N* = 572): Mean (SD), [N] if missing data, unless otherwise stated.**

| Characteristic | Children | | Adolescents | | Adults | | Total sample | |
|---|---|---|---|---|---|---|---|---|
| | ASD | TD | ASD | TD | ASD | TD | ASD | TD |
| *N* | 81 | 64 | 114 | 90 | 126 | 97 | 321 | 251 |
| Sex (Percentage male) | 70.37 | 60.94 | 76.32 | 68.89 | 69.84 | 72.16 | 72.27 | 68.13 |
| Age in years | 9.59 (1.50) | 9.52 (1.54) | 14.94 (1.71) | 15.39 (1.71) | 22.80 (3.55) | 23.25 (3.29) | 16.67 (5.92) | 16.93 (6.02) |
| Full-scale IQ | 105.54 (14.35) | 111.81 (12.50) | 101.81 (15.92) | 106.69 (13.32) | 103.97 (15.21) | 109.14 (12.29) | 103.60 (15.28) | 108.95 (12.82) |
| ADI-R RRB | 4.46 (2.89) [79] | - | 4.30 (2.68) [112] | - | 4.07 (2.54) [116] | - | 4.25 (2.68) [307] | - |
| RBS-R Stereotyped Behavior | 3.83 (3.33) [71] | 0.19 (0.68) [54] | 3.64 (3.97) [96] | 0.14 (0.62) [69] | 1.86 (2.92) [91] | - | 3.06 (0.16) [258] | 0.16 (0.63) [129] |
| RBS-R Ritualistic-Sameness | 7.48 (5.52) [71] | 0.35 (0.91) [54] | 7.39 (6.26) [96] | 0.41 (1.31) [69] | 4.79 (4.44) [91] | - | 6.50 (5.59) [258] | 0.36 (1.12) [129] |
| ADI-R Social Interaction | 15.14 (6.8) [79] | - | 17.46 (6.59) [112] | - | 14.78 (6.80) [116] | - | 15.85 (6.81) [307] | - |
| ADI-R Communication | 13.32 (5.56) [79] | - | 13.48 (5.56) [112] | - | 11.82 (5.67) [116] | - | 12.81 (5.64) [307] | - |
| SRS-2 SCI | 73.44 (11.19) [73] | 44.60 (5.10) [55] | 74.67 (10.89) [93] | 45.35 (6.05) [71] | 64.32 (10.89) [87] | - | 70.75 (11.90) [253] | 44.97 (5.58) [132] |
| ADHD hyper/impulsive parent-report | 4.33 (2.93) [72] | 0.37 (1.17) [52] | 2.77 (2.77) [96] | 0.20 (0.84) [71] | 1.33 (1.80) [94] | - | 2.68 (2.77) [262] | 0.25 (0.97) [130] |
| ADHD inattentive parent-report | 5.25 (3.00) [72] | 0.62 (1.60) [52] | 4.77 (3.12) [96] | 0.89 (1.81) [71] | 3.23 (3.20) [94] | - | 4.35 (3.22) [262] | 0.76 (1.70) [130] |
| ADHD hyper/impulsive self-report | - | - | - | - | 1.61 (1.99) [96] | 0.61 (1.43) [72] | 1.61 (1.99) [96] | 0.61 (1.43) [72] |
| ADHD inattentive self-report | - | - | - | - | 2.91 (2.38) [96] | 0.81 (1.51) [72] | 2.91 (2.38) [96] | 0.81 (1.51) [72] |
| Anxiety (BAI/BYI-II[a]) | 14.62 (8.77) [72] | 6.00 (4.97) [51] | 14.13 (10.05) [61] | 8.67 (7.05) [64] | 14.97 (13.24) [97] | 4.27 (5.1) [73] | - | - |
| Task behavior | | | | | | | | |
| Accuracy (overall) | 0.65 (0.11) | 0.68 (0.13) | 0.67 (0.13) | 0.76 (0.14) | 0.73 (0.15) | 0.77 (0.14) | 0.69 (0.14) | 0.74 (0.14) |
| PerErrors | 0.28 (0.14) | 0.26 (0.15) | 0.30 (0.18) | 0.23 (0.18) | 0.27 (0.20) | 0.21 (0.16) | 0.28 (0.18) | 0.23 (0.16) |
| Win-stay | 0.69 (0.16) | 0.70 (0.16) | 0.72 (0.16) | 0.81 (0.15) | 0.80 (0.16) | 0.84 (0.15) | 0.75 (0.17) | 0.79 (0.16) |
| Lose-shift | 0.55 (0.11) | 0.53 (0.14) | 0.50 (0.15) | 0.43 (0.17) | 0.45 (0.18) | 0.41 (0.19) | 0.49 (0.16) | 0.45 (0.18) |

[a] Parent-report for children, self-report for adults and adolescents.

Abbreviations: ADHD, attention-deficit hyperactivity disorder; ADI-R, Autism Diagnostic Interview-Revised; ASD, autism spectrum disorder; BAI, Beck Anxiety Inventory; BYI-II, Beck Youth Inventories–Second Edition; hyper/impulsive, hyperactivity/impulsivity; Lose-shift, changing the response following punishment as a proportion of total lose trials; PerErrors, perseverative errors, expressed as a proportion of reversal trials; RBS-R, Repetitive Behavior Scale-Revised; SD, standard deviation; SRS-2 SCI, Social Responsiveness Scale 2nd Edition Social Communication Index; TD, typical development; Win-stay, repeating the previous choice following reward expressed as a proportion of total win trials

(as a proportion of total rewarded trials) and changing the response following punishment (as a proportion of total punished trials). As in previous studies using this task [10, 47, 48, 50, 51], reaction time is not examined here because it is unlikely to capture task-relevant processes, since no response speed instructions are given nor is there a time limit for responding (see S1 Fig for further discussion).

## Reinforcement learning models

We compared three reinforcement learning models to examine different computational mechanisms driving information integration and the cognitive processes underlying learning and

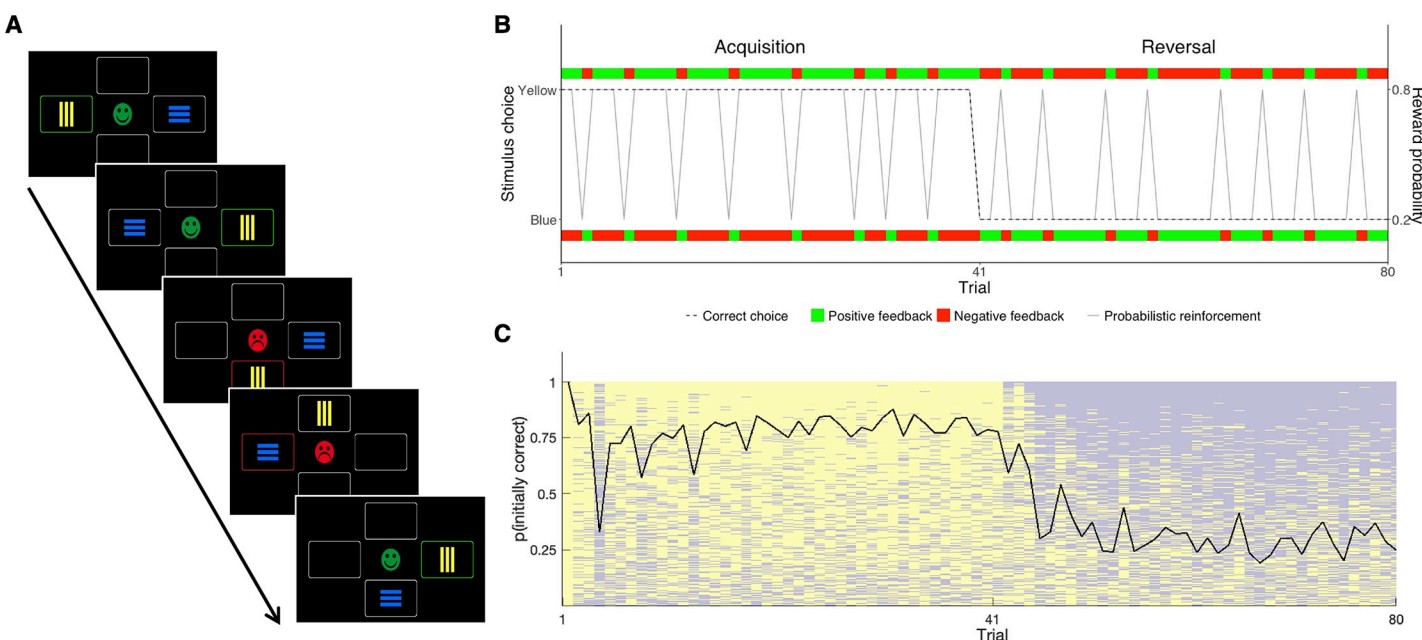

**Fig 1. Task presentation and pooled task behavior.** (A) An example of several consecutive trials—on each trial, participants have to choose between two stimuli, presented pseudorandomly in two of the four possible locations. Feedback is received in the form of a smiling green face (positive) or a sad red face (negative) and is probabilistic, meaning that some is "misleading" (e.g., trial 3). Win-stay trials are those in which individuals repeat their stimuli choice following positive feedback (e.g., trials 2 and 3), and lose-shift trials are those in which individual change their stimuli choice following negative feedback (e.g., trials 4 and 5). (B) The structure of the task —the first stimuli chosen by each participant is correct in the acquisition phase (trials 1–40; here: yellow). Feedback was given with an 80:20 reward/punishment ratio; green blocks indicate reward and red blocks indicate punishment. In the reversal phase (trials 41–80), the true correct stimulus is reversed (here: blue) as is the contingency schedule. (C) Overall trial-by-trial behavior—All participants' data, sorted by performance, with average performance overlaid (black line) regardless of diagnosis or age group. Compare to (B) to see how task structure is experienced in practice (see S1 Data).

flexible adaptation. Each model extends the Rescorla-Wagner value update rule [52] but in different ways in terms of how information is integrated. The Rescorla-Wagner update rule assumes that individuals assign and update internal stimulus value signals based on the prediction error, i.e., the mismatch between outcome (received reward/punishment following choice of this stimulus) and prediction (expected value of choosing this stimulus). Below, we omit results from the original Rescorla-Wagner model, as all other models consistently outperformed it (see S1 Text and S2 Table).

**(1) Counterfactual update model.** Previous studies suggest individuals may use counterfactual updating in reversal learning tasks, as it captures the anti-correlatedness of the choice stimuli (i.e., where one is correct, the other is incorrect; [53, 54]). The counterfactual update (CU) model extends the standard Rescorla-Wagner algorithm by updating the value of both choice stimuli.

$$V_{c,t} = V_{c,t-1} + \eta(O_{t-1} - V_{c,t-1}) \qquad (1)$$

$$V_{nc,t} = V_{nc,t-1} + \eta(-O_{t-1} - V_{nc,t-1}) \qquad (2)$$

Here, the value $V$ of both the chosen $c$ and unchosen $nc$ stimulus are updated with the actual prediction error and the counterfactual prediction error per trial $t$, respectively. $O$ is the outcome received. The learning rate $\eta$ evidences the magnitude of the value update affected by both prediction errors—put simply, the speed of learning. In this framework, reduced flexible behavior may be underpinned by too frequent response switches quantified by excessive value updating after punishment.

**(2) Reward-punishment model.** Alternatively, reduced flexible task behavior may result from reduced punishment learning. Reduced punishment learning would have a disproportionate effect during the reversal phase because punishments following choices of the previously rewarded stimulus would have a diminished influence on choice behavior due to a failure to devalue this stimulus. To assess whether this mechanism drives reduced flexible behavior, we use a different extension of the Rescorla-Wagner model, with separate learning rates for reward and punishment (reward-punishment model [R-P]; [47]). This allows for the capture of differential learning to feedback types.

$$V_{c,t} = \begin{cases} V_{c,t-1} + \eta^{rew}(O_{t-1} - V_{c,t-1}), \, if \, O_{t-1} > 0 \\ V_{c,t-1} + \eta^{pun}(O_{t-1} - V_{c,t-1}), \, if \, O_{t-1} < 0 \end{cases} \qquad (3)$$

Here, $\eta^{rew}$ is the learning rate for rewards and $\eta^{pun}$ is the learning rate for punishment; $O$ is the outcome received. In this model, only the chosen stimulus value is updated.

**(3) Experience-weighted attraction–dynamic learning rate model (EWA-DL).** Finally, reduced flexible behavior may result from a growing insensitivity to novel information. By this mechanism, a failure to update values based on new information (i.e., accumulating negative feedback denoting a true reversal) would cause perseveration of the previously rewarded response and delayed or even complete failure to switch. We examined this mechanism using the experience-weight parameter from a reduced version of the EWA model as presented in previous work [47], where we used the formulation of a nonstationary learning rate through updating of an experience weight. This dynamical learning rate allows for interpolation between different forms of updating (accumulating versus averaging rho shifts from 0 to 1). Note that we do not use the exact same model of the original EWA model [55], as we omit the feature of blending belief-based versus reinforcement learning. To make this distinction clear, we have labeled this model as EWA-DL (but note that it is the identical model to [47]). The EWA-DL model extends classic reinforcement learning with an experience-weight parameter that captures the attribution of significance to past experience over and above new information as an individual progresses through the task. This effectively reduces the learning rate over time. Thus, in this context, perseveration would arise from a slowness, after reversal, to update the value of the now usually rewarded stimuli due to an overreliance on preceding task experience. The growth of the experience weight $n$ and update of the stimulus values $V$ are defined as follows:

$$n_{c,t} = n_{c,t-1} \times \rho + 1 \qquad (4)$$

$$V_{c,t} = (V_{c,t-1} \times \varphi \times n_{c,t-1} + O_{t-1})/n_{c,t} \qquad (5)$$

Here, $n_{c,t}$ is the "experience weight" of the chosen stimulus on trial $t$, which is updated on every trial using the experience decay factor $\rho$. $V_{c,t}$ is the value of choice $c$ on trial $t$ for outcome $O$ received in response to that choice, and $\varphi$ is the decay factor for the previous payoffs. In this model, $\varphi$ is equivalent to the inverse of the learning rate in Rescorla-Wagner models (or alternatively, $n = 1 - \varphi$; see also [47]). For $\rho > 0$, the experience weights promote more sluggish updating with time. Previous work has shown the EWA-DL to be the winning model in neurotypical adults in the same PRL task [47].

**Softmax action selection.** For all models, a softmax choice function was used to compute the action probability given the action values. On each trial $t$, the action probability of choosing

option A (over B) was defined as follows:

$$p(A) = \frac{1}{1 + e^{\beta(\alpha - (V_A - V_B))}} \qquad (6)$$

Here, $\beta$ ($0 < \beta < 5$) is the inverse temperature parameter that governs the stochasticity of the choice, computed using inverse logit transfer. We set the upper bound to 5, as individual parameters are regularized by group-level parameters that prevent extreme parameter estimates (see parameter estimation section), and our data indeed showed that all $\beta$ estimates are smaller than 5. We refer to $\beta$ in this paper as value sensitivity, as it reflects sensitivity to the difference in stimulus values, that is, the degree to which a (perceived) difference in stimulus values determines choice (see S1 Text). Higher $\beta$ values denote decisions driven by relative value whereas lower $\beta$ values denote more choice stochasticity. Additionally, a small indifference point parameter $\alpha$ ($-0.5 < \alpha < 0.5$) is introduced, which captures any selection bias in which both options are equally likely to be selected. Including this indifference point parameter systematically improved performance of all models. The action probability of options A and B by definition sum to 1: $p(B) = 1 - p(A)$.

## Parameter estimation and model selection/validation

Parameter estimation was performed with hierarchical Bayesian analysis (HBA) using Stan language in R (RStan; [56, 57]), adopted from the hBayesDM package [58]. Posterior inference was performed using Markov chain Monte Carlo (MCMC) sampling in RStan. The models were fit separately for each of six groups—diagnosis (ASD, TD) × developmental stage (children, adolescents, adults)—and compared within each group to assess how well they fit the data (goodness-of-fit) while accounting for model complexity. Comparison of model fit was assessed per group using Bayesian bootstrap and model averaging, whereby log-likelihoods for each model were evaluated at the posterior simulations and a weight obtained for each model. Model weights include a penalizing term for model complexity and a normalizing term according to the number of models being compared; thus, for each group, model weights sum to 1 [59]. Higher model weight indicates better model fit. We conducted model recovery analyses, and, for completeness, we also ran model fitting across age groups (see S1 Text). Finally, we established that the winning models could replicate the observed behavior using one-step-ahead prediction (e.g., [60]). Here, parameters are drawn from the joint posterior distribution and combined with the outcome sequence to predict future choices thereby quantifying absolute model fit. That is, we let the model take random draws from each participant's joint posterior distribution to generate choices. We iterated this procedure as many times as the number of samples (i.e., 4,000) per trial per participant. We implemented two ways to assess posterior predictions. First, we computed the predictive accuracy using the number of correct predictions divided by the total number of iterations and tested if this accuracy was significantly better than chance level (i.e., 50%). Second, we analyzed the generated data in the same way as we analyzed the observed data and compared whether results from generated data captured the behavioral pattern in our behavioral analysis (for further details on model specification and validation, see S1 Text).

## Optimal learning parameters

We identified the optimal learning parameters for each model using simulation. Taking the CU model as an example, we first took the learning rate from a grid with 1,000 steps from 0 to 1 and then simulated choice data for every learning rate. We computed how often the simulated choice data matched the correct option (i.e., the more rewarding option). We repeated

this simulation 10,000 times and identified the optimal learning rate as the value that resulted in the highest choice accuracy. We used the same procedure to determine the optimal learning parameter(s) for the R-P model and the EWA-DL.

### Clinical measures

**ASD symptomatology.** Two measures were used to assess RRB symptom severity in ASD: (1) The ADI-R [44] is a structured parent/caregiver interview comprising 93 questions assessing most severe/early developmental ASD symptoms, which yields an algorithm score for RRB based on 12 items; (2) The Repetitive Behavior Scale-Revised (RBS-R; [61]) is a 43-item parent-report questionnaire tapping current RRB, which typically yields a total score and five subscales [62]. Here, we use the Ritualistic-Sameness and Stereotyped Behavior subscales as the best indices of behavioral rigidity (see S3 Table for a comparison of all subscales). To examine whether relationships were specific to RRB, ADI-R domain scores for Communication and Reciprocal Social Interaction were included, as were T-scores for the Social Communication Index on the Social Responsiveness Scale 2nd Edition (SRS-2; [63])—a parent-report questionnaire assessing current social-communication difficulties. On all measures, higher scores indicate greater symptom severity.

**Comorbid symptomatology.** The DSM-5 rating scale of ADHD [64] and the Beck Anxiety Inventory (BAI; [65]) were used to assess associated symptoms. For ADHD symptoms, parents of all ASD participants completed the parent-report form, and in addition, ASD adults completed the self-report form. For anxiety, adult participants completed the BAI in self-report form, whereas adolescents completed the self-report version of the anxiety subscale of the Beck Youth Inventories (BYI-II; [66]). Parents/caregivers of children completed the same BYI-II subscale in parent-report form.

### Statistical analysis

All analyses were conducted in R [67]. First, we characterized the cohort with respect to sex, age, and IQ differences. Second, to examine the effects of diagnosis and age group on the task performance measures, we employed linear mixed-effects models using the lme4 package in R [68]. The models included diagnosis and age group (and for accuracy, phase) as between-participant factors (including their interaction[s]) and site as a random factor. Including sex in the models did not improve model fit. Post hoc pairwise comparisons were computed from contrasts between factors using lsmeans package with Tukey adjustments [69]. Following the reinforcement learning model comparisons and validation using one-step-ahead predictions, we examined case-control differences on winning model parameters in each age group. Finally, we used correlational analyses to examine associations between task behavior, model parameters, and symptomatology. Symptomatology associations were conducted only in the ASD groups using Spearman's correlations owing to non-normality in scores. Significance thresholds for correlational analyses are Bonferroni-corrected for multiple comparisons—children/adolescents (.05/11): $p = .0045$ and adults (.05/13): $p = .0038$. Effect sizes are reported as Cohen's $d$.

## Results

### Sex, age, and IQ group differences

Diagnostic groups did not differ on sex or age, either overall or within each age group (all $p > .1$). However, all groups differed significantly on full-scale IQ, with TD groups scoring higher than ASD groups ($p$ ranging .01–.005; $d$ ranging 0.32–0.47). Therefore, for all further group

comparisons, we assessed whether results changed with IQ as a confound regressor, and, in addition, we conducted analyses of task behavior in an IQ-matched subsample (S2 Text and S4 Table). Results were largely unchanged throughout (see S2 Text and S2 Fig).

## Task behavior

Grouped trial-by-trial behavior is shown in Fig 2A and descriptive statistics in Table 1. All diagnostic and age groups performed above chance in both phases of the task, showing task comprehension (all $p < 2.2 \times 10^{-16}$; see S3 Text, S3 Fig and S5 Table). A repeated-measures analysis of accuracy showed significant main effects of phase ($F_{[1,566]} = 294.25$, $p < 2.2 \times 10^{-16}$), diagnosis ($F_{[1,566]} = 21.96$, $p = 9.52 \times 10^{-8}$), and age group ($F_{[2,566]} = 16.64$, $p = 3.49 \times 10^{-6}$) but no significant interactions (all $p > .1$). Post hoc analyses revealed accuracy was on average significantly higher (1) in the acquisition phase than in the reversal phase, reflecting the challenge of flexible adaptation ($p < .0001$, $d = 0.82$); (2) in TD individuals compared to ASD individuals ($p < .0001$, $d = 0.29$); and (3) in older age groups compared to younger age groups (adults-adolescents, $p = .0113$, $d = 0.22$; adults-children, $p < .0001$, $d = 0.51$; adolescents-children, $p = .0062$, $d = 0.29$; Fig 2B).

Next, a significant main effect of diagnosis on perseverative errors was observed ($F_{[1,565.42]} = 11.07$, $p = .0009$, $d = 0.30$; Fig 2C), such that ASD individuals made on average significantly more perseverative errors than TD individuals; however, there was no significant effect of age nor interaction between diagnosis and age group ($p > .2$). For both accuracy and perseverative

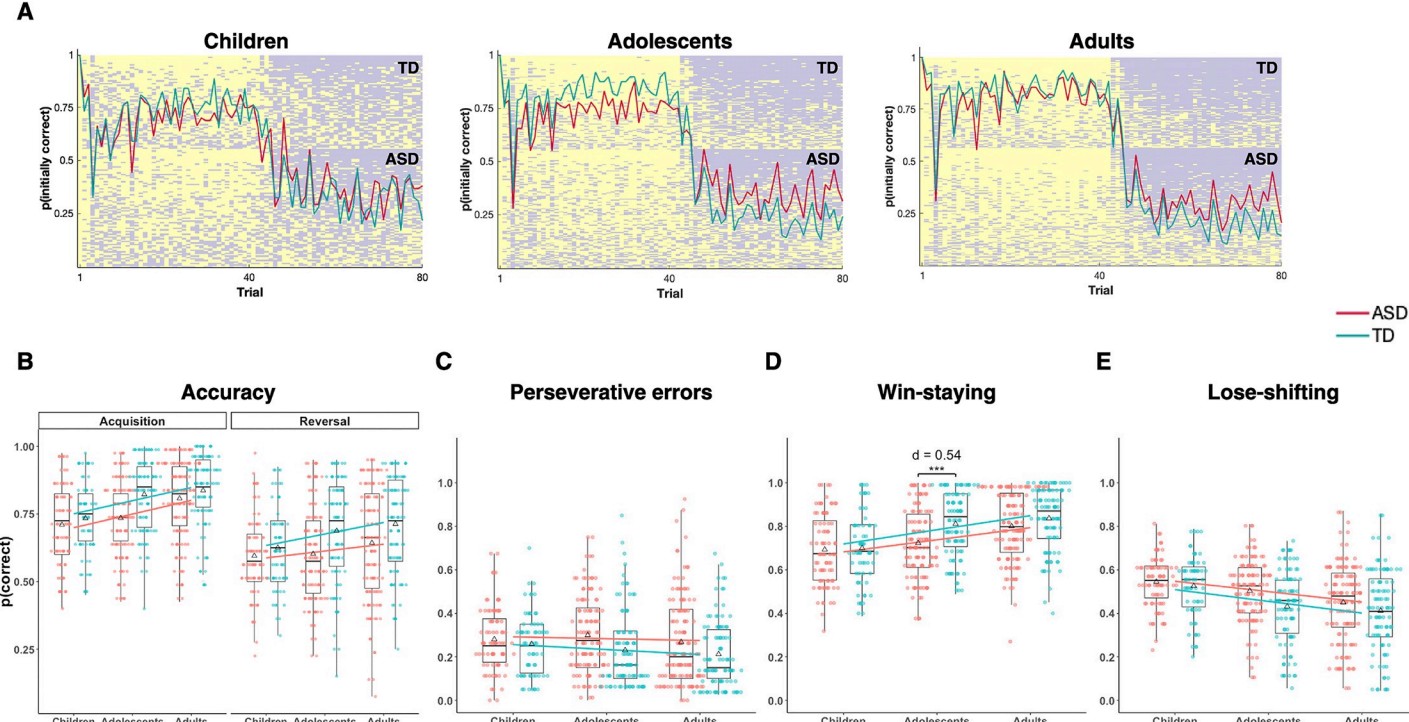

**Fig 2. Task behavior.** (A) Trial-by-trial data for each age group with diagnostic group averages overlaid. More evidence of task understanding in adults, as indicated by more correct task behavior and steeper shifts at reversal in comparison to children. (B) Task accuracy was greater (1) in the acquisition phase compared to the reversal phase, (2) in older age groups compared to younger, and (3) in TD individuals compared to ASD individuals. (C-E) Linear mixed-effects models showed a main effect of diagnosis for all three task performance measures (perseverative errors, win-staying, lose-shifting) and a main effect of age for win-staying (D) and lose-shifting (E) but not perseverative errors (C). For win-staying, a diagnosis × age group interaction was also found. Post hoc tests revealed ASD adolescents showed significantly reduced win-staying compared with TD adolescents (D), ***$p < .001$ (see S1 Data). ASD, autism spectrum disorder; TD, typical development.

errors, results were unchanged both in the IQ-matched subsample and with IQ as a confound regressor (S2 Text and S2 Fig).

Regarding feedback sensitivity, ASD individuals showed on average significantly less win-stay and more lose-shift behavior relative to TD individuals, and for both there was a main effect of age (win-stay: diagnosis [$F_{(1,563.28)}$ = 12.06, $p$ = .0006, $d$ = 0.24], age group [$F_{(2, 521.29)}$ = 27.78, $p$ = $3.4 \times 10^{-12}$]; lose-shift: diagnosis [$F_{(1, 564.28)}$ = 9.86, $p$ = .0018, $d$ = 0.23], age group [$F_{(2,390.88)}$ = 19.50, $p$ = $8.5 \times 10^{-9}$]). Pairwise post hoc comparisons revealed win-staying increased and lose-shifting decreased with age (Fig 2D and 2E). For win-stay behavior, the predicted interaction between diagnosis and age group was approaching significance ($p$ = .057). A between-diagnosis group analysis of each age group revealed ASD adolescents showed less win-staying than TD adolescents ($p$ < .0008; Fig 2D, $d$ = 0.54), which survived Bonferroni correction (correcting for task behavioral measures × age groups: $p$-value = .05/[3 × 3] = .0056). For lose-shift behavior, there was no significant interaction between diagnosis and age group ($p$ = .3). Results were again consistent in the IQ-matched subsample and when IQ was entered as a confound regressor (S2 Text and S2 Fig).

The pattern of results reported here is also replicated in the additional analyses conducted with the subset of ASD individuals who meet ADI-R criteria (S2 Text and S2 Fig).

## Model comparison and validation

Model weightings are shown in Fig 3A, and all winning model's parameters had independent contributions (S4 Fig). There were no between-diagnosis group differences in terms of model preference, only changes across development. Within both ASD and TD age groups, model weights showed that for children, the CU model provided the highest model evidence; for adolescents, the R-P model provided the highest model evidence; and for adults, the EWA-DL provided the highest model evidence. Results were unchanged when models were fitted with ($z$-scored) IQ as a covariate (see S6 Table). Model recovery results showed that all models' identities can be well recovered (S5 Fig). Collapsing age groups, the R-P model provided the highest model evidence in both diagnostic groups (S7 Table). One-step-ahead predictions of each group's winning model showed the models captured the key features of task behavior (e.g., the first response to negative feedback, the switch at reversal), with posterior predictive accuracy values of 0.61 and above. All models performed significantly better than chance level ($p \leq 1.23 \times 10^{-11}$). Average simulated behavior closely resembled participants' behavior (Fig 3B).

## Within-model diagnostic group comparisons

We then investigated which computational mechanisms underpin poorer task performance in ASD for the different age groups. To this end, we compared diagnostic groups on parameter estimates from the winning model of each age group (Table 2; see also S4 Text).

**Children—CU model.** ASD children showed a significantly higher learning rate than TD children ($t_{[140.46]}$ = 3.68, $p$ < .001, $d$ = 0.62; 95% confidence interval [CI] 0.26 to −0.93; Fig 3C). Simulations showed the optimal learning rate (i.e., leading to higher choice accuracy) for the CU model is 0.18 (Fig 3D, see also S1 Text), which is closer to the learning rate for TD children ($M_{TD}$ = 0.19) than the learning rate for ASD children ($M_{ASD}$ = 0.26). A higher learning rate in our learning schedule reflects oversensitivity to feedback (including probabilistic punishment, which should be ignored). There were no differences on the other model parameters ($\beta$, $\alpha$; $p$ > .1). Results were unchanged with IQ as a confound regressor.

**Adolescents—R-P model.** A repeated-measures feedback type × diagnosis linear mixed-effect model with learning rates as dependent variables showed a significant main effect of

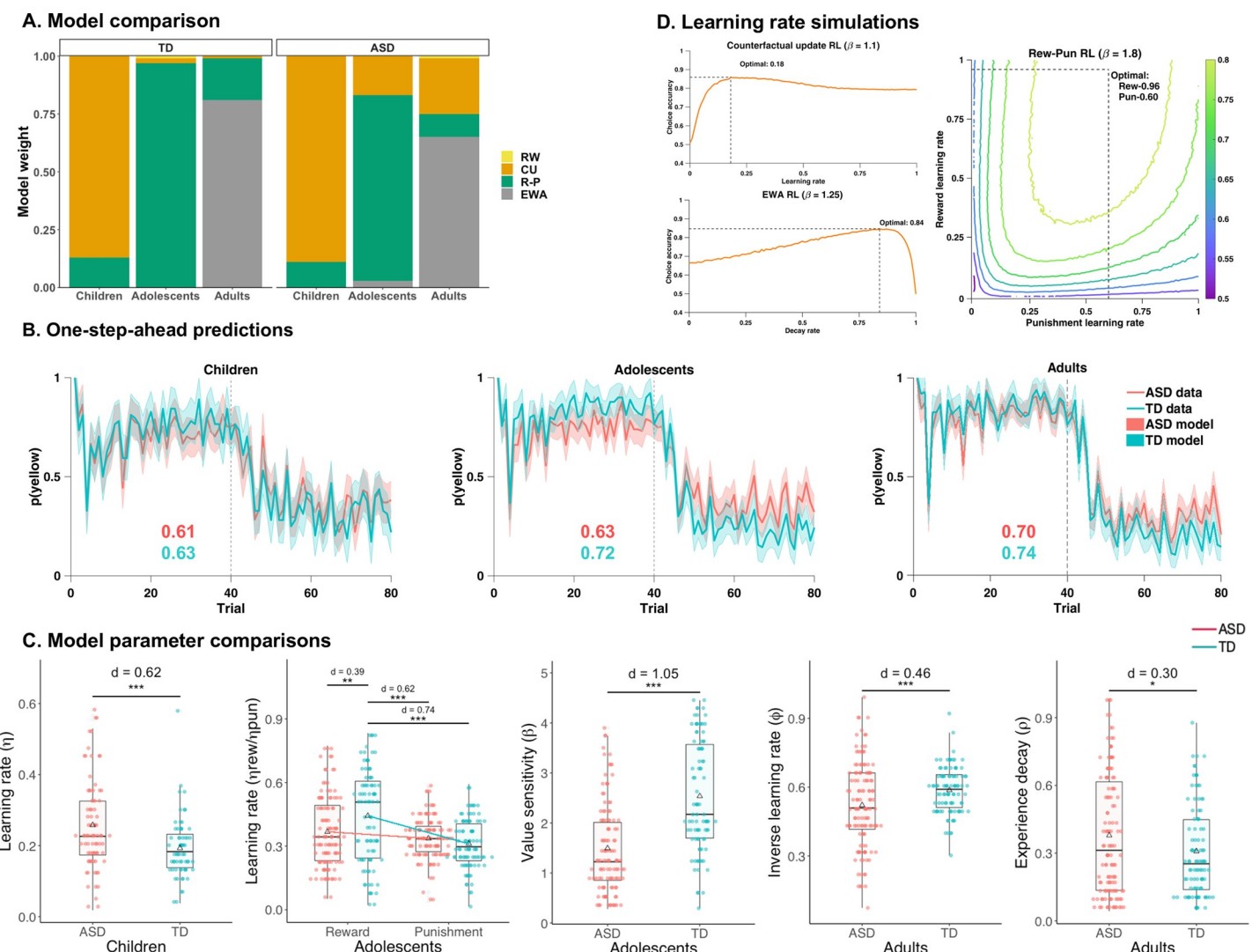

**Fig 3. Model comparisons, validations, and parameters.** (A) Evidence (model weights) for models within each diagnostic and age group. Very similar patterns are observed for TD and ASD groups; winning models for children, adolescents, and adults are the CU, R-P, and EWA-DL, respectively. (B) One-step-ahead posterior predictions for each age and diagnostic group according to winning models. Colored lines indicate diagnostic-group-averaged trial-by-trial task behavior; shaded areas indicate 95% HDI of the one-step-ahead simulation using the entire posterior distribution. Compare with actual task data in Fig 2A. Posterior predictive accuracies are also indicated on each plot (ASD: red; TD: blue). (C) Model parameter comparisons. Within each winning model and thus age group, parameter estimates were compared between diagnostic groups: (1) ASD children showed a significantly higher learning rate ($\eta$) than TD children, in which simulations showed the optimal learning rate to be 0.18; (2) ASD adolescents showed a significantly lower reward learning rate than TD adolescents, but no difference between punishment learning rates was observed; (3) ASD adults showed significantly lower $\varphi$ than TD adults, the optimal value was shown to be 0.85 in simulations, and ASD adults also showed significantly greater experience decay ($\rho$) than TD adults, suggesting great perseveration. (D) Learning rate simulations showing optimal learning rates for each model (Counterfactual update, compare to Fig 3C Children; Rew-Pun, compare to Fig 3C Adolescents—Learning rate; EWA, Experience-weighted attraction-dynamic learning rate, compare to Fig 3C Adults—Inverse learning rate). ***$p < .001$, **$p < .01$, *$p < .05$; $\Delta$ indicates group mean (see S1 Data). ASD, autism spectrum disorder; CU, counterfactual update; $d$, Cohen's $d$ model; EWA-DL, experience-weighted attraction–dynamic learning rate model; HDI, highest density interval; R-P, reward-punishment model; Rew-Pun, reward-punishment; RL, reinforcement learning; RW, Rescorla-Wagner; TD, typical development.

feedback type ($F_{[1,202]} = 33.04$, $p = 3.20 \times 10^{-8}$) and a significant interaction between feedback type and diagnosis ($F_{[1,202]} = 12.57$, $p = .0004$), but no main effect of diagnosis ($p = .1$; Fig 3C). Reward learning rates were significantly larger than punishment learning rates ($p < .0001$, $d = 0.43$). Pairwise post hoc comparisons showed autistic adolescents' reward learning rate was significantly lower than TD adolescents' reward learning rate ($p = .004$, $d = -0.39$), but their punishment learning rates were not significantly different ($p = .7$). Additionally, TD

**Table 2. Model parameters for each age and diagnosis group's winning model and within age-group comparisons.**

| | Mean (*SD*) | | Highest Density Interval (of MCMC) | | *d* | *p* value |
|---|---|---|---|---|---|---|
| | ASD | TD | ASD | TD | | |
| *Children – Counterfactual update* | | | | | | |
| $\eta$ | 0.258 (0.126) | 0.193 (0.087) | [0.206, 0.311] | [0.150, 0.235] | 0.600 | 0.0003 |
| $\beta$ | 0.979 (0.783) | 1.202 (0.892) | [0.886, 1.073] | [1.063, 1.340] | -0.266 | 0.117 |
| $\alpha$ | -0.014 (0.319) | -0.042 (0.153) | [-0.092, 0.069] | [-0.116, 0.025] | 0.114 | 0.482 |
| *Adolescents – Reward-punishment* | | | | | | |
| $\eta^{\mathrm{rew}}$ | 0.368 (0.169) | 0.443 (0.223) | [0.268, 0.466] | [0.359, 0.536] | -0.382 | 0.0039 |
| $\eta^{\mathrm{pun}}$ | 0.336 (0.098) | 0.311 (0.116) | [0.265, 0.402] | [0.264, 0.356] | 0.231 | 0.671 |
| $\beta$ | 1.494 (0.897) | 2.535 (1.108) | [1.290, 1.745] | [2.209, 2.854] | -1.033 | $1.51 \times 10^{-11}$ |
| $\alpha$ | -0.032 (0.255) | -0.031 (0.161) | [-0.088, 0.016] | [-0.076, 0.010] | -0.003 | 0.985 |
| *Adults – Experience-weighted attraction* | | | | | | |
| $\varphi$ | 0.521 (0.185) | 0.587 (0.101) | [0.476, 0.571] | [0.546. 0.630] | -0.439 | 0.0009 |
| $\rho$ | 0.379 (0.268) | 0.308 (0.200) | [0.292, 0.465] | [0.208, 0.407] | 0.298 | 0.026 |
| $\beta$ | 1.231 (0.742) | 1.290 (0.763) | [1.092, 1.378] | [1.131, 1.457] | -0.078 | 0.566 |
| $\alpha$ | -0.052 (0.308) | 0.040 (0.344) | [-0.120, 0.015] | [-0.030, 0.102] | -0.281 | 0.040 |

*SD* = standard deviation; MCMC = Markov Chain Monte Carlo sampling; *d* = Cohen's *d* effect size

adolescents' reward learning rate was significantly higher than both their punishment learning rate ($p < .001$, $d = 0.74$) and ASD adolescents' punishment learning rate ($p < .001$, $d = 0.62$).

In the context of the R-P model (with two learning rates), simulations showed the optimal reward and punishment learning rates for choice accuracy are 0.96 and 0.60, respectively (Fig 3D and S6 Fig). This optimal pattern of a reward learning rate higher than the related punishment learning rate is also shown in TD adolescents' learning rates, whereas autistic adolescents showed on average similar levels of reward and punishment learning and reduced learning from rewards compared to TD adolescents. In addition to reduced learning from rewards, autistic adolescents also showed significantly lower value sensitivity ($\beta$; $t_{[169.27]} = -7.24$, $p = 1.51 \times 10^{-11}$, $d = -1.05$, 95% CI $-1.32$ to $-0.73$), reflecting more stochastic choice behavior. These results suggest that reduced reward learning and lower value sensitivity drive worse task performance in ASD adolescents. Results were unchanged with IQ as a confound regressor.

**Adults—EWA-DL.** Autistic adults showed on average a significantly lower inverse learning rate ($\varphi$; $t_{[201.2]} = -3.37$, $p = .0009$, $d = -0.46$, 95% CI $-0.71$ to $-0.17$)—which is effectively comparable to a higher Rescorla-Wagner learning rate. Simulations show that in this model, the optimal value for $\varphi$ is 0.85 ($M_{\mathrm{ASD}} = 0.52$, $M_{\mathrm{TD}} = 0.59$; Fig 3D and S5 Fig). ASD adults also showed significantly higher experience-weight values ($\rho$) than TD adults ($t_{[220.82]} = 2.25$, $p = .021$, $d = 0.30$; 95% CI 0.04 to $-0.56$), indicating a faster reliance on past (acquisition) experience, leading to inflexibility. When IQ was entered as a confound regressor, the difference in $\varphi$ remained significant ($p = .004$), but the difference in experience decay ($\rho$) did not ($p = .2$).

For associations between task behavior and model parameters, see S4 Text and S8 Table.

## Symptomatology correlations in ASD

All correlations with symptomatology are listed in S9 Table and S10 Table. Here, we discuss only those that remained significant after Bonferroni correction for multiple comparisons.

In the ASD children, perseverative errors were positively correlated with anxiety (Fig 4A; $r_{72} = 0.34$, $p = .0040$). However, no associations with model parameters survived multiple

## Task behaviour

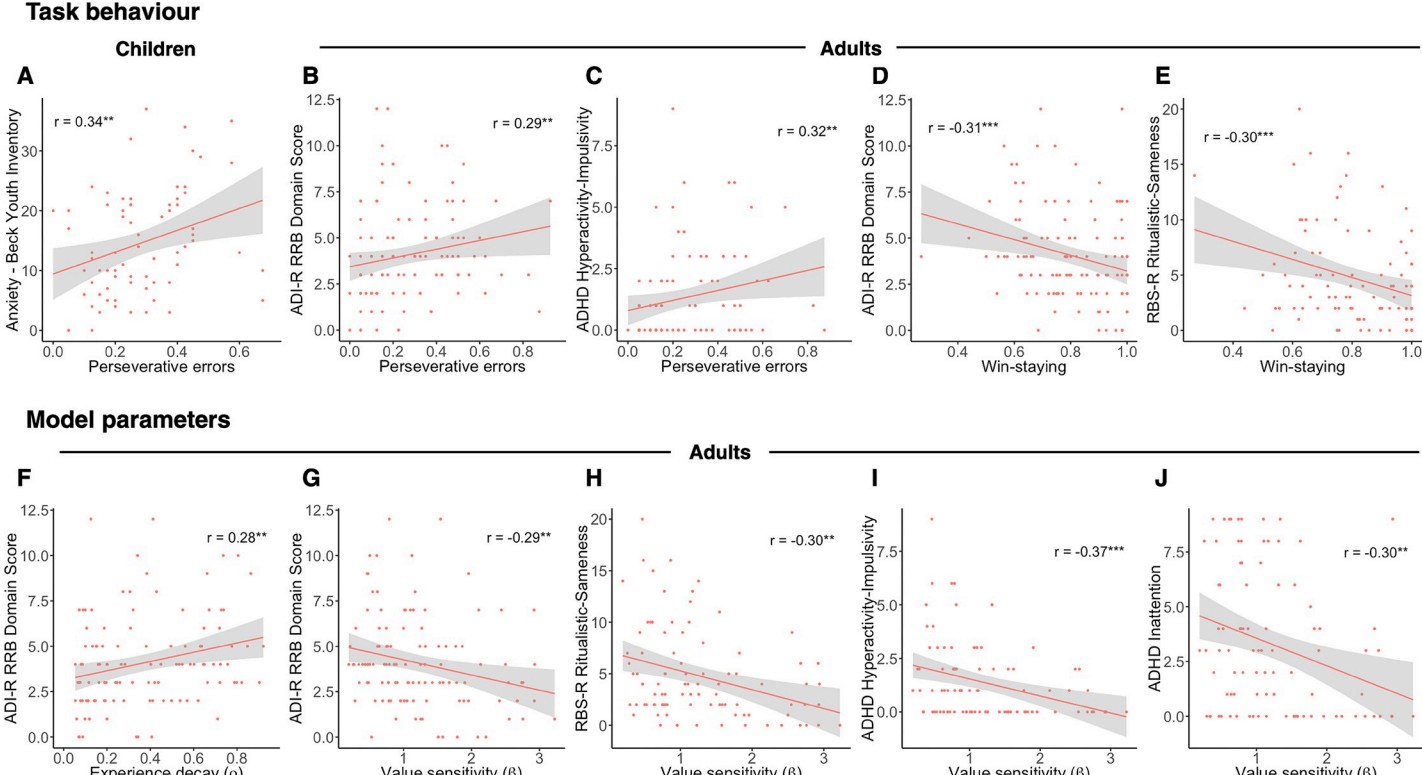

**Fig 4. Symptomatology correlations in ASD.** (A) In ASD children, perseverative errors were significantly correlated with anxiety ($r_{72}$ = 0.34, $p$ = .0040). In ASD adults, (B) perseverative errors were significantly correlated with ADI-R RRB ($r_{116}$ = 0.29, $p$ = .0013). (C) Perseverative errors were further significantly positively related to parent-reported ADHD Hyperactivity/Impulsivity ($r_{94}$ = 0.32, $p$ = .0017). Win-staying was significantly negatively related to (D) ADI-R RRB ($r_{116}$ = −0.31, $p$ = .0007) and (E) RBS-R Ritualistic-Sameness ($r_{91}$ = −0.30, $p$ = .0004). In ASD adults, experience decay ($\rho$) was significantly positively associated with (E) RRB (ADI-R RRB $r_{116}$ = 0.28, $p$ = .0022) as was (F, G) value sensitivity ($\beta$; ADI-R RRB $r_{116}$ = −0.29, $p$ = .0019; RBS-R $r_{91}$ = −0.30, $p$ = .0040). (H, I) Value sensitivity ($\beta$) was also significantly negatively correlated with parent-reported ADHD symptomatology (ADHD hyperactivity/impulsivity $r_{116}$ = −0.37, $p$ = .0003; ADHD inattention $r_{116}$ = −0.30, $p$ = .0037). ADHD, attention-deficit hyperactivity disorder; ADI-R, Autism Diagnostic Interview-Revised; ASD, autism spectrum disorder; RBS-R, Repetitive Behavior Scale-Revised; RRB, restricted, repetitive behavior (see S1 Data).

comparison corrections. For the adolescent group, neither associations with task behavioral measures nor model parameters survived Bonferroni correction. In the adult group, both perseverative errors and experience decay ($\rho$) were positively correlated with ADI-R RRB (perseverative errors–Fig 4B, $r_{116}$ = 0.29, $p$ = .0013; experience decay, $\rho$–Fig 4F, $r_{116}$ = 0.28, $p$ = .0022). Additionally, perseverative errors were positively associated with parent-reported ADHD hyperactivity/impulsivity (Fig 4C; $r_{94}$ = 0.32, $p$ = .0017), though this association would not survive Bonferroni correction when controlling for the RRB association ($r_{89}$ = 0.26, $p$ = .013). Win-stay behavior was negatively correlated with both ADI-R RRB and RBS-R Ritualistic-Sameness behavior (Fig 4D and 4E; ADI-R RRB $r_{116}$ = −0.31, $p$ = .0007; RBS-R Ritualistic-Sameness $r_{91}$ = −0.30, $p$ = .0004), and relatedly so was value sensitivity ($\beta$; Fig 4G and 4H; ADI-R RRB $r_{116}$ = −0.29, $p$ = .0019; RBS-R Ritualistic-Sameness $r_{91}$ = −0.32, $p$ = .0017). Value sensitivity was also negatively associated with parent-reported ADHD symptomatology in ASD adults (Fig 4I and 4J; ADHD hyperactivity/impulsivity $r_{116}$ = −0.37, $p$ = .0003; ADHD inattention $r_{116}$ = −0.30, $p$ = .0037).

No correlations with learning rates ($\eta$, $\eta^{rew}$, $\eta^{pun}$, $\varphi$) nor lose-shift behavior survived Bonferroni correction in any age group. Of note, no significant associations between either task behavior or model parameters and social-communication difficulties were observed.

## Discussion

In this study, we examined flexible behavior on a PRL task and used reinforcement learning models to investigate underlying learning mechanisms in autistic and neurotypical children, adolescents, and adults. Overall, we found evidence of on average reduced flexible behavior in autistic individuals, as indexed by poorer task performance across measures. Our results also show a developmental effect whereby older age groups outperformed younger age groups on the task. Using computational modeling of behavior, we showed that dominant learning mechanisms shift with developmental stage, but not diagnosis, and that poorer task performance in ASD is underpinned by atypical use of the age-related dominant learning mechanism in each age group. Furthermore, we found evidence for an association between perseveration and behavioral rigidity in ASD, but only in adults.

These findings emphasize the importance of a developmental framework when examining mechanistic accounts of both intact and reduced flexible behavior. Although the role of development is well documented in the neurotypical literature, particularly with respect to key brain regions for cognitive flexibility, goal-directed decision-making, and feedback learning [9, 26, 70], age-related differences in ASD have been relatively understudied. Examining learning mechanisms across development, we found dominant differential integration of reward and punishment feedback in both adolescent groups, corresponding with literature that suggests neurotypical adolescents are hyperresponsive to rewards [29, 71]. In contrast, children's behavior was best captured by a single learning rate, and adults showed evidence of increasingly weighting their accumulating experience to inform subsequent decisions and slow down new learning. This dominant experience-weight mechanism in adults is consistent with previous neurotypical research [47]; however, our study is the first to report the same dominant mechanism in ASD adults. These results therefore posit that cognitive and reinforcement-based processes are governed primarily by age, leading to the relative dominance of different learning mechanisms in different age groups. In this way, differential feedback learning may be developing in children and strengthened in adolescence, and experience weighting may similarly develop and then prevail in adulthood.

Previous research suggests that reversal learning—and, more broadly, cognitive flexibility—is impaired in ASD (e.g., [1, 72]) and may be underpinned by the recruitment of different brain regions to TD [22]. Our findings provide support for the impairment hypothesis in that on average the ASD group was less accurate and more perseverative and showed reduced outcome sensitivity compared to the TD group. Furthermore, this pattern of results was consistent in both subsample analyses, showing robustness of findings in both an IQ-matched subsample and a subsample including only those ASD individuals who reach ADI-R criteria [46]. Notably, autistic adolescents showed reduced win-staying compared to TD adolescents, in line with previous studies that showed reduced win-staying in adults [21, 22]. However, in this study, we did not find reduced win-staying specifically in autistic adults compared to TD adults.

Our computational modeling findings suggest that reduced flexible behavior in the ASD group is underpinned by significant differences in the efficient use of learning mechanisms within each age group on this task. Both the children and adult ASD groups showed faster learning rates compared to their TD counterparts. Here, faster learning rates are less optimal, as they result in reduced ability to ignore probabilistic feedback. These results are consistent with predictive coding and Bayesian accounts of ASD that suggest "overlearning" in response to feedback and difficulties ignoring noise, putatively due to precise or inflexible prediction errors [37, 38]. Indeed, studies using volatile task environments or near-chance reward contingencies have reported intact learning and updating or superior performance in ASD [22, 39]. In these contexts, fast learning rates are optimal, as changes are more frequent and therefore updating must be too.

Thus, findings demonstrate that altered learning rates in ASD have different effects on behavior depending on the learning environment and, in tandem, that computational models characterize differences rather than solely deficits, shedding light on environments in which differences may be expressed as strengths rather than difficulties. The computational differences in ASD appear to manifest as pronounced difficulties when the environment is less volatile, and learning when to ignore probabilistic feedback is as important as tracking change. These difficulties may underpin the marked difficulties with minor (probabilistic) deviations in routines or unexpected changes in ASD that caregivers so frequently report [73]. In different environments, faster learning may manifest in strengths; these differences have important implications for intervention development.

In ASD adolescents, reduced flexible behavior—and, particularly, reduced win-staying—was underpinned by reduced reward learning compared to TD adolescents. This finding is consistent with previous research showing impaired reward circuitry dysfunction in autistic adolescents [74]. Whereas neurotypical adolescents are thought to demonstrate increased risk due to high reward sensitivity, reduced reward learning in autistic adolescents may result in reduced risk-taking and serve as a protective effect [75]. Reduced reward learning could also have implications for behavioral interventions. If autistic adolescents do not learn from typical rewards in the same way that TD adolescents do, the type(s) of rewards used in behavioral interventions would require adapting [76]. For example, there is evidence to suggest autistic individuals assign specific reward value to their circumscribed interests such that they may be of value in intervention design [77–79].

Reduced flexible behavior has previously been associated with RRB in ASD [1, 80–82], though results are not consistent despite a strong theoretical link. Here, we observed robust, moderately strong associations between perseveration and RRB in autistic adults. We also found no evidence of associations with social-communication difficulties, providing support for the specificity to RRB. On the RBS-R, these associations were specific to the Ritualistic-Sameness and Stereotyped Behavior subscales, capturing behavioral rigidities. Previous literature has also reported associations between flexibility impairments and RRB symptom severity in ASD adults [83] with mixed findings in children and adolescents [82, 84–86]. Moving forward, examining this association across developmental stages will continue to be important.

To our knowledge, this study is the first to elucidate a potential learning mechanism by which behavioral rigidity manifests in autistic adults: perseveration as a result of a reluctance or inability to switch—"getting stuck"—because new information is devalued in favor of past experience, which in turn impedes updating choice behavior. Furthermore, as this mechanism has been associated with dopamine transporter differences in neurotypical adults [47], and abnormalities in the dopaminergic system have been implicated in ASD [87], this study highlights a potential mechanistic link between neurobiology and behavior worthy of further study.

Beyond perseveration, RRB in autistic adults positively associated with reduced value sensitivity (i.e., more stochastic choice behavior). This mechanism was also associated with more ADHD symptoms in autistic adults. Reduced value sensitivity has previously been identified as a key factor in poor task performance in anhedonia [88]. Together, these findings suggest that value sensitivity may have transdiagnostic value in explaining aspects of reduced flexible behavior. As altered decision-making is prevalent across many neurodevelopmental and neuropsychiatric disorders, examining underlying processes in relation to symptom dimensions rather than purely diagnostic categories will likely be of greater value for understanding implicated brain circuitries [89].

In autistic adolescents, we found no relationship between performance measures or learning mechanisms and clinical symptoms. In children with ASD, we observed a positive

association between perseverative behavior and anxiety symptoms. Previous studies have demonstrated a relationship between anxiety and reduced flexible behavior in non-autistic adults [90, 91] and children and adolescents with anxiety disorders [92]. One plausible link between perseveration and anxiety may be the intolerance of uncertainty (IU) construct, as uncertainty is inherent in probabilistic tasks. IU is a core construct in anxiety disorders [93] and a possible transdiagnostic mechanism [94] shown to be relevant for anxiety in ASD [95]. Associations between anxiety and RRB in ASD have frequently been reported [96, 97]. Together, our findings broadly support the notion that reduced flexible behavior is of clinical relevance in ASD; however, the extent to which particular processes may be differentially linked to specific aspects of RRB versus commonly co-occurring features of anxiety or ADHD at different developmental stages will require further examination.

## Limitations

This study has a number of limitations. Firstly, despite the large sample size and wide age range, the sample does not include children younger than 6 or adults above 30 years of age. Future research including very young children and older adults could allow for the assessment of any other age-related changes in dominant learning mechanisms. Secondly, it is important to note that each group's winning model is only relative to the other models tested here—although we note that the models capture behavior well and perform far above chance. However, it is (always) possible that other models may perform even better and further models may be developed in the future. A full model with all parameters combined was not possible because of convergence issues, emphasizing the relative dominance of learning mechanisms rather than any suggestions of mutual exclusivity. We highlight, nevertheless, that the study is the first to compare reinforcement learning models in ASD across age groups. Thirdly, our approach necessitated that we implicitly treated each diagnostic and age group as relatively homogeneous. The increasing recognition of the considerable phenotypic and etiological diversity of ASD indicates potential individual differences in learning processes within or across these a priori defined subgroups. Estimating the learning strategy for each individual would allow for a "bottom-up" approach to identifying potential subgroups based on learning strategies. Fourth, our sample was limited to individuals with an ASD diagnosis and TD counterparts. Given that reduced flexible behavior and atypical reinforcement learning are implicated in many other areas of psychiatry, it would be informative to extend this study with a transdiagnostic sample, in the context of the research domain criteria framework (RDoC; [89]). Additionally, given the growing literature suggesting differential reward processing in ASD, future work could assess potential differences in learning and flexible behavior in the context of different reward modalities, i.e., use different types of feedback, such as monetary stimuli. Finally, it will be crucial to verify our results through replication. The current sample has been reassessed as part of a longitudinal project, thereby providing some opportunity for this.

## Conclusions

Current results suggest group-level impairments in flexible behavior across developmental stages in ASD. We show evidence of developmental shifts in dominant computational mechanisms underlying PRL that are consistent across ASD and TD individuals. Within each age group, differences in model parameter estimates showed less optimal learning in ASD, underpinning poorer task performance. Additionally, we show that perseverative behavior—and, in adults, learning mechanisms—were related to behavioral rigidities or co-occurring symptoms of anxiety or ADHD. Findings emphasize the importance of understanding reduced flexible

behavior in ASD within a developmental framework and underline the strength of computational approaches in ASD research.

## Supporting information

**S1 Data. Excel spreadsheet containing, in separate sheets, the underlying numerical data for figures and figure panels: 1C, 2A-2E, 3C, 3D, 4A-4J, S1, S2A-S2L, S3A-S3B, S4, and S7.** (XLSX)

**S1 Text. Supplementary methods.** (DOCX)

**S2 Text. Additional IQ and subsample analyses.** (DOCX)

**S3 Text. Evidence of learning.** (DOCX)

**S4 Text. Further results for comparisons of model parameter estimates.** (DOCX)

**S1 Fig. z-RTs in the PRL task averaged across task trials; shaded area represents the standard deviation.** Notably, reaction times do not change at the point following reversal, illustrating that reaction times are unlikely to reflect task-relevant processes. PRL, probabilistic reversal learning; z-RT, reaction time (z-scored). (TIF)

**S2 Fig. Box plots showing task behavior for (A-D) the full sample, (E-H) the IQ-matched subsample, and (I-L) Risi and colleagues' ADI-R criteria ASD subsample.** The pattern of results remains largely unchanged across both subsample analyses. ADI-R, Autism Diagnostic Interview-Revised; ASD, autism spectrum disorder. (TIF)

**S3 Fig. Evidence of learning.** (A) Trial-by-trial average proportion of correct responses (here, yellow in acquisition phase, blue in reversal phase) plotted separately for the groups that passed and failed the learning criterion. The red lines indicate the mean for that task phase (acquisiton/reversal) and the orange lines indicate the 95% confidence intervals. Thus, both groups performed above chance in both task phases. (B) Diagnostic and age group average proportion of correct responses for each task phase, plotted separately for the pass/fail groups to confirm that perfgormance above chance was maintained even within diagnostic and age subgroups. (TIF)

**S4 Fig. Independent contribution of model parameters.** Pair plots of each group's winning model parameters for ASD (top panel) and TD (bottom panel). In each pair plot, diagonal plots show marginal distributions of each parameter; off-diagonal plots show pairwise scatters of parameters. ASD, autism spectrum disorder; CU, counterfactual update model; EWA, experience-weighted attraction–dynamic learning rate model; RP, reward-punishment model; TD, typical development. (TIF)

**S5 Fig. Model recovery.** Data from 40 synthetic participants were simulated with each of our three main models. Color indicates model weights calculated with Bayesian model averaging using Bayesian bootstrap (higher model weight value indicates higher probability of the candidate model to have generated the observed data). CU, counterfactual update model; EWA,

experience-weighted attraction–dynamic learning rate model; RP, reward-punishment model.
(TIF)

**S6 Fig. Simulation showing a larger value difference for a higher reward learning rate (TD) than a lower reward learning rate (ASD), when punishment learning rates are comparable.** ASD, autism spectrum disorder; TD, typical development.
(TIF)

**S7 Fig. Highly correlated factual and counterfactual learning rates.**
(TIF)

**S1 Table. Participant numbers and ADI-R scores (mean, SD) for the full ASD sample and Risi and colleagues' (2006) ADI-R criteria subsample.** ADI-R, Autism Diagnostic Interview-Revised; ASD, autism spectrum disorder; SD, standard deviation.
(DOCX)

**S2 Table. Effective number of parameters for the RW and CU models.** CU, counterfactual update; RW, Rescorla-Wagner.
(DOCX)

**S3 Table. Behavior and model parameter estimates correlations with all RBS-R subscales.** RBS-R, Repetitive Behavior Scale-Revised.
(DOCX)

**S4 Table. Descriptive statistics (mean, SD—unless otherwise stated) for the full sample and the IQ-m, within age and diagnostic groups, with $p$-values for within-age group, between diagnostic group comparisons of age, sex, and IQ.** IQ-m, IQ-matched subsample; SD, standard deviation.
(DOCX)

**S5 Table. Numbers, proportions, and chi-squared statistics for learning criterion attainment status (pass/fail) by diagnostic and age groups.**
(DOCX)

**S6 Table. Model weights for model runs with IQ as a covariate.**
(DOCX)

**S7 Table. Model weights for model runs with age groups collapsed.**
(DOCX)

**S8 Table. Correlations between task behavior and model parameters.**
(DOCX)

**S9 Table. Correlations between task behavior, age, IQ, and symptomatology.**
(DOCX)

**S10 Table. Correlations between model parameters, age, IQ, and symptomatology.**
(DOCX)

## Acknowledgments

We thank all participants and their families for their efforts to participate in the study. We also acknowledge the contributions of the whole EU-AIMS LEAP group: Sara Ambrosino, Bonnie Auyeung, Tobias Banaschewski, Simon Baron-Cohen, Sarah Baumeister, Christian F. Beckmann, Christian Beckmann, Sven Bölte, Thomas Bourgeron, Carsten Bours, Michael

Brammer, Daniel Brandeis, Claudia Brogna, Yvette de Bruijn, Bhismadev Chakrabarti, Ineke Cornelissen, Flavio Dell'Acqua, Guillaume Dumas, Sarah Durston, Christine Ecker, Claire Ellis, Jessica Faulkner, Vincent Frouin, Pilar Garcés, David Goyard, Lindsay Ham, Hannah Hayward, Joerg Hipp, Rosemary Holt, Mark H. Johnson, Prantik Kundu, Meng-Chuan Lai, Xavier Liogier D'ardhuy, Michael Lombardo, David J. Lythgoe, René Mandl, Luke Mason, Maarten Mennes, Andreas Meyer Lindenberg, Carolin Moessnang, Nico Mueller, Laurence O'Dwyer, Marianne Oldehinkel, Bob Oranje, Gahan Pandina, Antonio M. Persico, Barbara Ruggeri, Amber Ruigrok, Jessica Sabet, Roberto Sacco, Emily Simonoff, Will Spooren, Julian Tillmann, Roberto Toro, Heike Tost, Jack Waldman, Steve C. R. Williams, Caroline Wool- dridge, and Marcel P. Zwiers.

## Author Contributions

**Conceptualization:** Daisy Crawley, Lei Zhang.

**Data curation:** Daisy Crawley, Lei Zhang, Hanneke den Ouden.

**Formal analysis:** Daisy Crawley, Lei Zhang.

**Funding acquisition:** Emily J. H. Jones, Tony Charman, Jan K. Buitelaar, Declan G. M. Mur- phy, Eva Loth.

**Investigation:** Daisy Crawley, Jumana Ahmad, Bethany Oakley, Antonia San José Cáceres.

**Methodology:** Daisy Crawley, Lei Zhang, Christopher Chatham, Hanneke den Ouden, Eva Loth.

**Project administration:** Eva Loth.

**Resources:** Declan G. M. Murphy, Hanneke den Ouden, Eva Loth.

**Software:** Lei Zhang, Hanneke den Ouden.

**Supervision:** Emily J. H. Jones, Christopher Chatham, Hanneke den Ouden, Eva Loth.

**Visualization:** Daisy Crawley, Lei Zhang.

**Writing – original draft:** Daisy Crawley, Lei Zhang.

**Writing – review & editing:** Daisy Crawley, Lei Zhang, Emily J. H. Jones, Jumana Ahmad, Bethany Oakley, Antonia San José Cáceres, Tony Charman, Jan K. Buitelaar, Declan G. M. Murphy, Christopher Chatham, Hanneke den Ouden, Eva Loth.

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
