## [Editor Report · Decision Letter 0]

12 Dec 2019

Dear Dr Crawley, 

Thank you for submitting your manuscript entitled "Modeling flexible behavior in children, adolescents and adults with autism spectrum disorder and typical development" for consideration as a Research Article by PLOS Biology.

Your manuscript has now been evaluated by the PLOS Biology editorial staff, as well as by an Academic Editor with relevant expertise, and I am writing to let you know that we would like to send your submission out for external peer review.

Please re-submit your manuscript within two working days, i.e. by Dec 16 2019 11:59PM.

Kind regards,

Gabriel Gasque, Ph.D.,

Senior Editor

PLOS Biology

---

## [Decision Letter · Decision Letter 1]

24 Jan 2020

Dear Dr Crawley,

Thank you very much for submitting your manuscript "Modeling flexible behavior in children, adolescents and adults with autism spectrum disorder and typical development" for consideration as a Research Article at PLOS Biology. Your manuscript has been evaluated by the PLOS Biology editors, by an Academic Editor with relevant expertise, and by three independent reviewers. You will note that reviewer 2, Stefano Palminteri, has signed his comments.

In light of the reviews (below), we will not be able to accept the current version of the manuscript, but we would welcome re-submission of a much-revised version that takes into account the reviewers' comments. We cannot make any decision about publication until we have seen the revised manuscript and your response to the reviewers' comments. Your revised manuscript is also likely to be sent for further evaluation by the reviewers.

We expect to receive your revised manuscript within 2 months. 

**IMPORTANT - SUBMITTING YOUR REVISION**

Your revisions should address the specific points made by each reviewer. Please pay special attention to the comments reviewers 1 and 3 raise regarding IQ matching. We think it is fundamental to address this point fully for a successful revision. We also think reviewer 2 raises sophisticated points about the models, which should be thoroughly addressed.

Please submit the following files along with your revised manuscript:

*Re-submission Checklist*

*Published Peer Review*

*PLOS Data Policy*

*Blot and Gel Data Policy*

Sincerely,

Gabriel Gasque, Ph.D., 

Senior Editor

PLOS Biology

REVIEWS:

Reviewer #1: This is a revised manuscript, I did not see the original. The authors describe results from a study of flexible behavior in a large sample of children, adolescents, and adults with autism spectrum disorder (ASD). The study uses computational modeling to quantify latent variables indexing perseveration and feedback sensitivity, and assess these variables in relation to diagnosis, developmental stage, and symptoms of ASD. In the participants completing the probabilistic reversal learning task, the authors observe that those with ASD showed more perseveration and less feedback sensitivity than typically developing (TD) peers, and older individuals in both groups showed greater feedback sensitivity than younger individuals. The computational modeling results suggest that learning mechanisms underlying flexible behavior differ across developmental stages, and in those with ASD reduced flexible behavior was driven by less optimal learning. Different aspects of task performance were related to symptoms in children vs. adults with ASD. This is an interesting, well-powered study on a timely topic. Some suggestions are below.

It is interesting that the authors choose to use a probabilistic reversal learning (PRL) task to study flexibility in autism, rather than a traditional cognitive flexibility task. This is because the PRL task includes a reward outcome. Thus, both flexibility and reward processing are indexed by such a task, and reward processing may also be altered in ASD. Can the authors elaborate in the Introduction on their decision to use such a task, rather than some variant of the Wisconsin Card Sort Task or other task-switching or set-shifting paradigms that do not involve a reward component? 

Is there previous literature to indicate that the specific reward/punishment stimuli used in this task are appropriately sensitive for indexing reward/punishment in both ASD and TD groups across age? 

Why was reaction time not assessed in addition to accuracy? It would be helpful to include this information as well as statistics comparing groups. 

It is very surprising that the authors do not attempt to match the ASD and TD groups on full-scale IQ. It would be best to take a subset of the larger sample that is matched on full-scale IQ to see if the results obtained from the larger sample still hold. It is not sufficient to simply include IQ as a confounding regressor. 

The authors may wish to see the following relevant work: 

The paradox of cognitive flexibility in autism. Geurts HM, Corbett B, Solomon M. Trends Cogn Sci. 2009 Feb;13(2):74-82.

Demystifying cognitive flexibility: Implications for clinical and developmental neuroscience. Dajani DR, Uddin LQ. Trends Neurosci. 2015 Sep;38(9):571-8.

Reviewer #2, Stefano Palminteri: Daisy and colleagues presents a study of reinforcement learning across development, contrasting neurotypical and ASD subjects. The authors report that both factors (age and psychiatric scores) affect behavioral performance in this probabilistic reversal learning task (in particular metrics of efficiency and feedback sensitivity). They then apply computational modeling to this task and report that, across different age groups, choices are not explained by the same computational model and that some parameters differ across different age groups. There is much to like about this paper, such as the big sample size (not usual for this kind of research), coupling development and ASD research with computational modeling. The paper has the potential to become a mile-stone in the field. I do have however some important concerns about the model specifications, selection and statistical analyses. 

FP model specification

One of the models is labelled 'fictitious play' (FP) and consists in a Rescorla-Wagner model where the value of the unchosen option is also updated (with the opposite prediction error). The first (but not the main) issue with this model is that the name if wrong and misleading. In behavioural game theory 'fictitious play' refers to a way of finding a best response to an opponent play by iteratively mentally simulating the other player response (see chapter 'Learning' of Camerer's book): a situation that clearly does not apply here, as FP learning concerns beliefs, not values. The more important issue concerning this model is that, in its current specification, the factual and counterfactual updates are governed by the same learning rate. This is problematic as the functional complexity (it has an additional equation compared to the Rescorla-Wagner) of the model is not quantified by an additional parameter. This is also problematic as it does not allow to quantify the counterfactual update separately from the factual one. It would be psychologically plausible to suppose that the two are differentially affected by ASD and age. 

EWA model specification

Another model is the one labelled 'Experience-Weighted Attraction' model. I have also an issue concerning this labeling. Again, the EWA model has been developed for game theory not bandits; as this paper is addressed to psychologists and neuroscientists, rather than game theorists, I find the labeling unfortunate. Furthermore the key distinctive feature of the EWA model is counterfactual update (i.e., update of the value of the unchosen strategy, based on the opponent's choice), which is not relevant here. If anything the current model il closer to Erev and Roth's model (American Economic Review, 1995) than the EWA. Yet, labeling is not the main issue I have with this model. In this paper the EWA model embodies an interesting (and plausible) idea, that is, when "experience" increases (i.e., number of trials per option) new outcomes count less. The problem is that it does so in a formalism that is quite different compared to that used in the other two models. For instance, while for some sets of parameter the EWA model can approximate the RW model, for many other parameters values option values will converge to very different quantities. Furthermore, the fact that experience weights can increase unboundedly (when the number of trials increases), is at odds with neurobiologically plausible instantiations of RL (see any basal ganglia model, for instance). To obviate this issues and ensure commensurability with the other models, the authors should replace the EWA model with Miller's model (Psychol Rev. 2019), which suppose a parallel 'habit' learning (equivalent to the experience weight) with a formalise that is closer to that of the other models and parameters that are psychologically easy to interpret. 

Softmax specification

The authors included a bias term in the softmax. I have issue with this parameter as I suspect it is capturing part of the effects induced by less ad hoc processes such as learning asymmetric, counterfactual update and increased habits. I suspect it's value is consistent with a bias toward the first rewarded stimulus. Am I correct? In any case, I would not include it, unless it is strongly justified by model comparison and model falsification. 

Model space specification

I think the model space should include the RW model + the full model that include all the features (counterfactual update learning rate; two learning rates for positive and negative prediction errors and the habit learning system of Miller) as two "extreme" benchmarks (hypo n vs. hyper-parameterisation). 

Model comparison and selection

I really liked the Bayesian model selection approach and the fact that authors showed also the simulation. However it would be interesting to see how the models perform on the other behavioral metrics (perseveration errors, win/stay etc.) to have a better idea of what behavioural feature is falsifying the "losing" models. It would also be important to show the model recovery on simulated datasets (i.e., the capacity to retrieve the correct model by model comparison in data where the ground truth is known). Finally, it would be interesting to know which is the "overall" winning model (across all ages). 

Statistics on the parameters 

I guess the parameters are correlated between them (to some extent, it is always the case). It would be interesting to verify the parameter recovery to check their correct estimability (similar to model recovery). Maybe a structural equation modeling (SEM)- approach would be useful to assess the effect of clinical score while controlling for the correlation between parameters and between scores. I also suspect that in the population, there are subjects whose behavior is random (i.e., the likelihood of any model model is not different from assuming random choices). It would be interesting to check the correlations (or SEM results on the non-random subjects (as parameters values of random subjects do not make sense). Finally, once the "full" model implemented, it would be interesting to perform the correlations (or SEM) on the parameters of the full model. 

Discussion beyond ASD

I know that it is not central in this study, but, as I believe that authors should mention that fact that that while some studies (we found it: Lefebvre et al 2019; others too: Van Slooten et al, 2018, 2019) showed higher learning rates for positive compared to negative learning rates, some others (Niv et al 2011; Gershman, 2015) found the opposite. As the present study contributes to this debate, it would be worth mentioning the "controversy" in the discussion. 

Line 157 

Please avoid the term "poor" in describing adolescent decision-making. 

Reviewer #3: 

Thank you for the opportunity of reviewing the manuscript « Modeling flexible behavior in children, adolescents and adults with autism spectrum disorder and typical development » and apologies for the delay with which I am sending my review. 

Restricted and repetitive behaviours are key symptoms in ASD. These symptoms are plausibly linked to diminished flexibility, which can be measured and modeled using standard reverse learning tasks. In these tasks, participants must learn contingency rules and adjust to shifts in these rules in response to negative feedback. Some amount of perseveration is warranted (it would be maladaptive to change our minds every time we get a single negative feedback - 100% contingencies are rare in life) but too much perseveration is often the signature of poor cognitive flexibility. In this paper, the authors test large sample sizes (572 children, adolescents and 102 adults with ASD; N=321 and typical development; TD; N=251) using such a reversal learning task. Their results show that ASD participants perseverate more on average than TD participants, which is consistent with prior findings in the literature. They also find that feedback sensitivity is modulated by age, which is also consistent with prior findings in the literature. Finally, they find that the learning rule used by individuals with ASD is less optimal than the rule used by individuals in the control group, and that these differences relate to measurable symptoms (anxiety and perseveration). 

Does the paper address a question of sufficiently broad interest ? 

The paper addresses a question that is of broad interest: Restricted and repetitive behaviours are key symptoms in ASD, which is a common disorder affecting millions of people. But beyond ASD, understanding the development of learning is an important and broad question. 

Does the paper make a sufficient leap in the literature ? 

Much of what the paper shows has been argued for in the literature. But the paper makes a leap by testing a large sample, using empirical methods that are robust and solid computational analyses. Autism research has been fraught with papers using small sample sizes and poorly characterized samples, which makes a lot of the literature essentially uninterpretable. This paper, by contrast, is a strong, and much needed contribution. As the authors state in the introduction, the current literature is inconsistent and one of the main reason for this is poor methods, including small samples sizes. 

Is the analysis correct ? 

The task is sound and the analysis appears correct to me but I have strong reservations about sample characteristics, especially in the absence of a pre-registration plan. It is obviously too late to pre-register but given the value of this hard-to-acquire data, I think that this is a major mistake. It is great that a culture shift is happening towards increased sample sizes, but this should be combined with a culture shift towards better research practices overall, including pre-registration. I do not think that this is grounds for rejection (it would be punitive!) but I do think that it is a great shame that we have no way of knowing how much of the analysis was planned ahead of time, what was decided a priori, and what should be considered exploratory. This is all the more concerning that there are many degrees of freedom when dealing with such data (much clinical data was collected, many models were tested, many dependent variables can be compared). 

Here are some examples: 

1) I was surprised not to see the ADOS (and only the ADI-R) - was the ADOS collected ? I did not see information allowing me to know whether all participants in the sample were above overall ADIR cut-off for an ASD, or above SRS cut-off. If not, have the autors tried to restrict their analysis to the subsample that meets all diagnostic criteria? 

2) How much were the correlations between performance and symptom severity predicted ? Is it a problem that associations are found for some clinical outcomes but not others (even though some of them arguably tap the same construct). 

3) Based on the methods section, it isn't clear to me how the ASD and TD group were matched. Do they *happen* to not differ on age and sex or were they *chosen* to match on these variables ? Given that the groups differ on IQ, I was wondering why the authors had not decided to restrict their analysis to a well-matched sample. 

I do not think that the additional analyses provided in the SM address this concern appropriately. Covariance analyses assume statistical properties that are hard to meet and the analyses are hard to interpret when the groups differ substantially on the covariate, which they do. These analyses also require that the relationship between covariate and outcome be the same in both groups (ie regression slopes), which they are not. (unless I missed something)

One option would be to test a larger control group (because it is comparatively harder to gather data from patients). This would allow the authors to regress performance in the RL task against IQ in the TD group and then check for each individual in the ASD group whether there is a discrepancy between observed and expeected performance given observed IQ. 

If that is not feasible in the current study, I would recommend random matching on age, sex and IQ (using an automated algorithm such as the matching package in R) and the subsequent analyses to be presented in the main text. 

There are indeed issues with matching but given the particular topic at stake here and given the statistical assumptions behind covariate analyses, this strategy appears sounder statistically. Specifically, matching may be better recommended when the variable used to match groups really controls for a constraint on experimental task performance that isn't central to the hypothesis. My understanding is that we are in a such a situation here IQ is a predictor of learning but the hypothesis is about perseveration, not intelligence.

At the very least, given that the analyses were not pre-registered, the reader should have the option of seeing what happens when multivariate matching is done (on age sex and IQ). If the results are strikingly different, this should be discussed transparently.

---

## [Decision Letter · Decision Letter 2]

21 Aug 2020

Dear Dr Crawley,

Thank you for submitting your revised Research Article entitled "Modeling flexible behavior in children, adolescents and adults with autism spectrum disorder and typical development" for publication in PLOS Biology. I have now obtained advice from the original reviewers 1 and 2 and have discussed their comments with the Academic Editor, who also assessed the way you answered to the concerns originally raised by reviewer 3.

Based on the reviews, we will probably accept this manuscript for publication, assuming that you will modify the manuscript to address the remaining points raised by reviewer 2. Please also make sure to address the data and other policy-related requests noted at the end of this email.

We expect to receive your revised manuscript within two weeks. 

***IMPORTANT:

Your revisions should address the specific points made by reviewer 2. In addition, we would like you to consider changing your title, which we think is too descriptive. We recommend one that conveys the central biological message and suggest the following. However, we are open to discuss alternatives:

“Modeling flexible behavior deficits in Autism Spectrum Disorder shows age-dependency and less optimal learning within each age group.”

Please submit the following files along with your revised manuscript:

In addition to the remaining revisions and before we will be able to formally accept your manuscript and consider it "in press", we also need to ensure that your article conforms to our guidelines. A member of our team will be in touch shortly with a set of requests. As we can't proceed until these requirements are met, your swift response will help prevent delays to publication.

*Copyediting*

*Published Peer Review History*

*Early Version*

*Submitting Your Revision*

Sincerely,

Gabriel Gasque, Ph.D.,

Senior Editor,

ggasque@plos.org,

PLOS Biology

ETHICS STATEMENT:

-- Please include the full name of the IACUC/ethics committee that reviewed and approved the animal care and use protocol/permit/project license. Please also include an approval number.

-- Please include the specific national or international regulations/guidelines to which your animal care and use protocol adhered. Please note that institutional or accreditation organization guidelines (such as AAALAC) do not meet this requirement.

-- Please include information about the form of consent (written/oral) given for research involving human participants. All research involving human participants must have been approved by the authors' Institutional Review Board (IRB) or an equivalent committee, and all clinical investigation must have been conducted according to the principles expressed in the Declaration of Helsinki.

DATA POLICY:

We note that you have stated in the online submission system that readers can access the raw data by contacting the EU-AIMS LEAP group. However, we request that you provide the underlying numerical values that underlie the summary data displayed in the following figure panels: Figures 1C, 2ABCDE, 3CD, 4A-J, S1, S2A-L, S3AB, S4, and S7.

The numerical data provided should include all replicates AND the way in which the plotted mean and errors were derived (it should not present only the mean/average values).

For an example see here: http://www.plosbiology.org/article/info%3Adoi%2F10.1371%2Fjournal.pbio.1001908#s5

These data can be made available in one of the following forms:

Please also ensure that each figure legend in your manuscript include information on where the underlying data can be found, and ensure your supplemental data file/s has a legend.

Reviewer remarks:

Reviewer #1: The authors have done a very nice job conducting new analyses to address reviewer comments. 

Reviewer #2: Overall, I think that the authors did a good job in addressing my concerns. The additional clarifications and analyses are very welcome. As I mentioned in my previous review, I think this paper would become an important milestone in the computational psychiatry of ASD. I do, however, have few remaining questions. 

1/ In response to my point R2.1 (counterfactual learning rate issue) the authors mention that (quote): 

"we would like to point out that our method of model comparison does indeed account for this increased functional complexity in the counterfactual update model, unlike simpler methods based on e.g. AIC/BIC comparison. This is one of the motivations from our perspective to conduct all computational modeling analyses in the Bayesian framework, where inferences are drawn from joint posterior distributions, rather than point estimates. Under the Bayesian framework, the joint parameter spaces of the CU and the simple RL model differ; thus, the effective number of parameters of these two models is different. It is the latter (i.e., effective number of parameters) that we use in the penalizing term in computing the model evidence"

However, it is still unclear to me how it is possible that functional complexity (i.e., adding one additional equation - or complexifying it) is taken into account for penalization in their Bayesian approach. As this is a crucial issue (and their claim is counterintuitive), I believe that the authors should unpack and explain this point better in the revised manuscript. 

2/ In response to my point R2.2 the authors decided to maintenant the name EWA for their model, based on the fact that they used the same name in a previous paper (2013), while they acknowledge that their model lacks the distinctive features of the EWA model. I still think that the label is not appropriate and, in a sense, "historically unfair". I strongly encourage the authors to take a look at Table 6.2 and table 6.3 of the book "Behavioral Game Theory" (by Colin Camerer), they will see their model will be classified as "Reinforcement Learning" (Roth and Erev) by Camerer himself.

Let's say a first group invent a model with the process X inside and call it 'John'. A second group invent a model with the processes X and Y inside and call it 'Paul'. A third group uses a model with the process X inside and call it 'Paul with no Y'. The first group would have all the rights to be annoyed, right? Of course this is not such a big deal, but I would love to know what are the authors' thoughts on this. 

3/ in the SI 'Additional methods' I believe the authors wanted to cite Lefebvre 2017, rather than 2018.

---

## [Editor Report · Decision Letter 3]

22 Sep 2020

Dear Dr Crawley,

On behalf of my colleagues and the Academic Editor, Franck Ramus, I am pleased to inform you that we will be delighted to publish your Research Article in PLOS Biology. 

Early Version

PRESS 

Kind regards,

Alice Musson

Publishing Editor, 

PLOS Biology

on behalf of

Gabriel Gasque,

Senior Editor

PLOS Biology